# Sexual Functioning in Female Patients Undergoing Surgical Treatment for Colorectal Cancer—A Single-Center, Prospective Triple Timepoint Yearly Follow-Up

**Magdalena Tarkowska [1,\*], Iwona Głowacka-Mrotek [2], Bartosz Skonieczny [3], Tomasz Nowikiewicz [3], Michał Tarkowski [4], Michał Jankowski [3], Wojciech Zegarski [3] and Piotr Jarzemski [1]**

[1] Department of Urology, Nicolaus Copernicus University in Toruń, Collegium Medicum in Bydgoszcz, 85-094 Bydgoszcz, Poland; piotr.jarzemski@cm.umk.pl

[2] Department of Rehabilitation, Nicolaus Copernicus University in Toruń, Collegium Medicum in Bydgoszcz, 85-094 Bydgoszcz, Poland; iwona.glowacka@cm.umk.pl

[3] Department of Surgical Oncology, Nicolaus Copernicus University in Toruń, Collegium Medicum in Bydgoszcz, 85-094 Bydgoszcz, Poland; skonieczny@gumed.edu.pl (B.S.); tomasz.nowikiewicz@cm.umk.pl (T.N.); jankowskim@cm.umk.pl (M.J.); zegarskiw@cm.umk.pl (W.Z.)

[4] Department of Security and Crisis Management, Cuiavian-Pomeranian Voivodeship Office in Bydgoszcz, Ludwik Rydygier Regional Polyclinical Hospital in Toruń, 87-100 Toruń, Poland; e.michal@wp.pl

\* Correspondence: m.tarkowska@cm.umk.pl; Tel.: +48-3655-306

**Abstract:** The study was aimed at assessing the quality of sexual functioning in female patients having undergone surgical treatment for cancer depending on the type of surgery. The prospective cohort consisted of 48 female patients (23 patients with stoma [A2] and 25 patients with maintained continuity of the GI tract [A1]). Study methods included a diagnostic survey and the analysis of medical records of patients. Research tools consisted of a standardized FSFI questionnaire and a proprietary form for evaluation of sociodemographic data. Measurements were performed at three timepoints: On the day before the surgery (Measurement I) as well as six and 12 months after the surgery (Measurements II and III, respectively). Statistically significant differences in results were observed in Measurements II and III in the subscales of arousal (II:$p = 0.0068$, III:$p = 0.0018$), lubrication (II:$p = 0.0221$, III:$p = 0.0134$), orgasm (II:$p = 0.0044$, III:$p = 0.0021$), satisfaction (II:$p = 0.0021$, III:$p = 0.0433$), and pain/discomfort (II:$p = 0.0343$, III:$p = 0.0473$). In all cases, lower scores corresponding to lower quality of sexual functioning were observed in patients in whom stoma had been performed. Statistically significant differences in sexual functioning were observed at Measurements II and III in each group, with the results being significantly ($p > 0.05$) worse in patients having undergone Hartmann's procedure or abdominoperineal resection). Variables significantly affecting self-assessed sexual satisfaction included marital status, age, and modality of neoadjuvant treatment. Restoration of the continuity of the gastrointestinal tract is a chance for better self-assessment of the patient's quality of life as regards sexual functioning.

**Keywords:** colorectal cancer; stoma; sexual functioning; quality of life

## 1. Introduction

Despite the implementation of screening programs, colorectal cancer is one of the most common malignancies in both male and female patients in developed countries [1]. Female patients diagnosed with colorectal cancer and subjected to surgical and systemic treatment are at risk of numerous sexual dysfunctions [2]. Literature data suggest that such dysfunctions may affect up to 19–62% of women with this diagnosis [3]. A literature review showed that 30–40% of patients who were sexually active before treatment became sexually inactive after treatment [4].The main reasons responsible for sexual activity being limited or discontinued in these patients include dyspareunia, vaginal dryness, and reduced libido. Surgical treatment of colorectal cancer frequently requires formation of a

temporary or permanent stoma, which undoubtedly changes the perception of one's own body in the context of sexual attractiveness. Treatment for colorectal cancer may involve surgery, chemotherapy, and/or radiation. Delivering treatment near the genital organs can negatively affect the function of the female sex organs [5].

The altered perception of one's own body, the stage of the neoplastic process, neoadjuvant and adjuvant treatment modalities, and frequently also limited ability to perform one's social role significantly contribute to intensification of psychosocial components responsible for reduced libido [2]. The aim of this study was to perform a prospective, single-center assessment of factors affecting the quality of sexual life in women having undergone surgeries for colorectal cancer within a one-year follow-up period.

## 2. Material and Methods

The study was designed as a single-center, prospective, triple timepoint pre-test post-test observation. The conduct of the study was approved by the Bioethics Committee at the Nicolaus Copernicus University in Toruń (decision no. 283/2019). The study was conducted at the Clinical Department of Oncological Surgery of the Franciszek Łukaszczyk Oncology Centre in Bydgoszcz. The group included in statistical analysis consisted of 48 patients having undergone colorectal cancer surgeries by means of anterior resection (either open or laparoscopic), Hartmann's resection, or abdominoperineal resection methods. Stoma was performed in 23 patients (group A2), whereas the continuity of gastrointestinal tract was maintained in another 25 patients (group A1). The study was conducted from June 2019 through August 2021. The quality of patients' sexual life was assessed at threetimepoints: Measurement I was performed before the surgical intervention, Measurement II (CATI) was performed six months after the surgery, and Measurement III (CATI) was performed 12 months after the surgery. Due to the restrictions resulting from the spread of COVID-19 in Poland, the first phase of study recruitment lasted from June 2019 to March 2020 and was followed by a two-month break until recruitment was continued in July 2020 and August 2020.

The inclusion criteria were as follows:

- good overall health status (Eastern Cooperative Oncology Group [ECOG] score of 0–1);
- voluntary, written consent to participate in the study,
- hospitalization at the Clinical Department of Oncological Surgery of the Franciszek Łukaszczyk Oncology Centre in Bydgoszcz at the time of recruitment;
- no distant metastases;
- age of up to 70 years;
- patients married or staying in partnership for at least 12 months prior to the surgery.

The exclusion criteria were as follows:

- class 3 obesity (Body Mass Index of >40);
- concomitance of other serious diseases (>ASA II);
- TNM (tumor, nodes, metastases) stage IV disease;
- continuity of the digestive tract being restored during the study.

The study was conducted using the diagnostic survey method. A proprietary questionnaire was used to collect demographic data on the patient sample, namely information on patients' age, educational background, area of residence, employment status, parity, socioeconomic status, and marital status.

Sexual satisfaction was assessed using the Female Sexual Function Index (FSFI) questionnaire. The use of the tool was authorized by its developers. The FSFI questionnaire is an international standardized tool to assess the quality of sexual life in women. It consists of 19 questions comprising a total of six domains, including desire, arousal, lubrication, orgasm, satisfaction, and pain/discomfort. The result is interpreted on the basis of total scores obtained in individual subscales; the higher the score, the better the quality of

individual sexual functioning-related components. The questions relate to the latest four weeks of the respondent's life.

Medical records of patients were analyzed to obtain relevant clinical data. Patients' weight, height, BMI, type of surgery, modality of neoadjuvant treatment, modality of adjuvant treatment, duration of hospital stay, incidence of postoperative complications, and TMN tumor staging were determined for the purposes of statistical analysis.

At the first phase of the study, a total of 107 anterior resections, 88 laparoscopic anterior resections, 33 Hartmann's procedures, and 56 abdominoperineal resections were performed at the Clinical Department of Oncological Surgery of the Oncology Centre in Bydgoszcz. The inclusion criteria were met by 65 patients who had been operated on. A total of 17 patients withdrew from the study at individual time points, were lost to follow-up, or provided incomplete answers to the survey questions. Thus, all study phases (June 2019–August 2021) were completed by a total of 48 patients. It's shown on Figure 1.

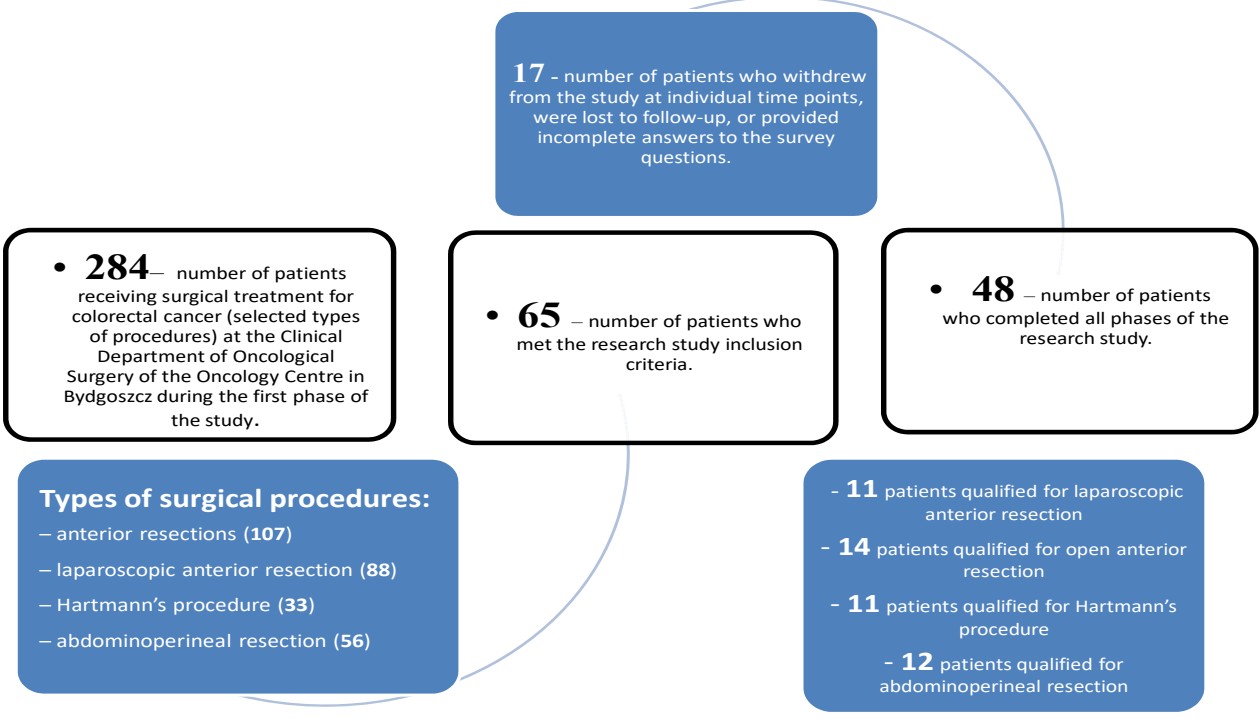

**Figure 1.** Scheme describing the exclusion of patients from the study.

Statistical analyses were carried out using the PQStat statistical package version 1.8.2.188. The weight, height and BMI values were compared between the groups (no stoma [A1] vs. stoma [A2] using Student's *t*-test. The duration of hospital stay was compared between the groups (no stoma [A1] vs. stoma [A2] using Mann–Whitney's U-test. Demographic and medical data were compared between the groups (no stoma [A1] vs. stoma [A2] using the chi$^2$ of exact Fisher tests (depending on the Cochrane condition being/not being met). FSFI scores were compared between the groups (no stoma [A1] vs. stoma [A2] using Mann–Whitney's U-test. FSFI scores at individual measurement timepoints were analyzed using the Friedman's test and the Dunn–Bonferroni post hoc test. Correlations between FSFI scores and the quantitative demographic variables were analyzed by estimation of Spearman's rank coefficients. Correlations between FSFI scores and the qualitative demographic variables were compared using Mann–Whitney's U-test (k = 2) or the Kruskall-Wallis test (k > 2) as well as the Dunn–Bonferroni post hoc test.

Test probability of $p < 0.05$ was defined as statistically significant, whereas test probability of $p < 0.01$ was defined as highly significant.

## 3. Results

The compared groups groups (no stoma [A1] vs. stoma [A2] were characterized in terms of clinical and sociodemographic parameters. High significance ($p < 0.0001$) was observed for the differences between the groups in terms of the type of the surgical procedure. With regard to group A1, 56% of patients were qualified for anterior resection of rectum while the remaining 44% were qualified for laparoscopic anterior resection of rectum. In group A2, 52% of patients were subjected to abdominoperineal resection of rectum while the remaining 48% were subjected to the Hartmann's procedure. High significance ($p = 0.0004$) was observed for the differences between the groups in terms of the type of neoadjuvant treatment. Qualification for neoadjuvant treatment was more common in patients in whom stoma was performed. Induction radiotherapy was the modality of choice in most patients within group A2, whereas radiochemotherapy was the most common neoadjuvant modality within group A1. With regard to demographic variables, both groups differed significantly ($p = 0.0301$) in terms of marital status. A higher percentage of married patients was observed in the no stoma group as compared to the stoma group. No significant differences were observed between the compared groups in terms of the type of adjuvant treatment ($p = 0.6623$), incidence of postoperative complications ($p = 0.7195$), tumor TNM staging ($p = 0.7717$), age ($p = 0.9869$), educational background ($p = 0.6834$), area of residence ($p = 0.6163$), employment status ($p = 0.7369$), parity ($p = 0.7051$), socioeconomic status ($p = 0.6812$), weight ($p = 0.5037$), height ($p = 0.8405$), BMI ($p = 0.4411$) and duration of hospital stay ($p = 0.7353$). Details are presented in Tables 1 and 2.

**Table 1.** Clinical and sociodemographic data.

| Variable | | No Stoma (A1) | | Stoma (A2) | | chi$^2$/Fisher's Test ($p$) |
|---|---|---|---|---|---|---|
| | | **N** | **%** | **N** | **%** | |
| Type of surgery | Anterior resection | 14 | 56% | 0 | 0% | <0.0001 |
| | Laparoscopic anterior resection | 11 | 44% | 0 | 0% | |
| | Abdominoperineal resection | 0 | 0% | 12 | 52.17% | |
| | Hartmann's procedure | 0 | 0% | 11 | 47.83% | |
| Neoadjuvant treatment | none | 9 | 36% | 0 | 0% | 0.0004 |
| | chemotherapy | 0 | 0% | 3 | 13.04% | |
| | radiochemotherapy | 11 | 44% | 7 | 30.43% | |
| | radiotherapy | 5 | 20% | 13 | 56.52% | |
| Adjuvant treatment | none | 17 | 68% | 15 | 65.22% | 0.6623 |
| | chemotherapy | 7 | 28% | 5 | 21.74% | |
| | radiochemotherapy | 1 | 4% | 3 | 13.04% | |
| Post-surgical complications | none | 21 | 84% | 18 | 78.26% | 0.7195 |
| | observed | 4 | 16% | 5 | 21.74% | |
| TNM stage of the disease (I–III) | I | 9 | 36% | 7 | 30.43% | 0.7717 |
| | IIA | 8 | 32% | 11 | 47.83% | |
| | IIB | 1 | 4% | 0 | 0% | |
| | IIIB | 3 | 12% | 3 | 13.04% | |
| | IIIC | 4 | 16% | 2 | 8.7% | |

**Table 1.** *Cont.*

| Variable | | No Stoma (A1) | | Stoma (A2) | | chi²/Fisher's Test (*p*) |
|---|---|---|---|---|---|---|
| | | **N** | **%** | **N** | **%** | |
| Educational background | elementary | 2 | 8% | 4 | 17.39% | 0.6834 |
| | vocational | 5 | 20% | 3 | 13.04% | |
| | secondary | 12 | 48% | 12 | 52.17% | |
| | higher | 6 | 24% | 4 | 17.39% | |
| Area of residence | urban | 19 | 76% | 16 | 69.57% | 0.6163 |
| | rural | 6 | 24% | 7 | 30.43% | |
| Employment status | retired/on disablement pension | 20 | 80% | 16 | 69.57% | 0.7369 |
| | regular employment/company owner | 4 | 16% | 6 | 26.09% | |
| | housekeeping | 1 | 4% | 1 | 4.35% | |
| Parity | 0 | 1 | 4% | 2 | 8.7% | 0.7051 |
| | 1 | 3 | 12% | 2 | 8.7% | |
| | 2 | 15 | 60% | 12 | 52.17% | |
| | 3 | 4 | 16% | 4 | 17.39% | |
| | 4 | 1 | 4% | 1 | 4.35% | |
| | 5 | 0 | 0% | 2 | 8.7% | |
| | 7 | 1 | 4% | 0 | 0% | |
| Marital status | married | 22 | 88% | 14 | 60.87% | 0.0301 |
| | in partnership | 3 | 12% | 9 | 39.13% | |
| Socioeconomic status | very good | 1 | 4% | 1 | 4.35% | 0.6812 |
| | good | 12 | 48% | 8 | 34.78% | |
| | average | 11 | 44% | 14 | 60.87% | |
| | low | 1 | 4% | 0 | 0% | |

*p*—significance level, A1—no stoma, A2—stoma.

**Table 2.** Weight, height, BMI, hospitalization time, and age.

| | Weight | | Height | | BMI | | Hospitalization Time | | Age | |
|---|---|---|---|---|---|---|---|---|---|---|
| | **A1** | **A2** | **A1** | **A2** | **A** | **A2** | **A1** | **A2** | **A1** | **A2** |
| M | 69.42 | 66.46 | 159.6 | 159.24 | 27.22 | 25.86 | 7.72 | 8.39 | 62.48 | 62.43 |
| Me | 66.6 | 65 | 160 | 158 | 25.07 | 26.04 | 7 | 7 | 65 | 66 |
| SD | 15.94 | 14.38 | 6.76 | 5.45 | 5.95 | 6.21 | 4.73 | 5.96 | 9.3234 | 9.5908 |
| Mann–Whitney's U-test/Student's *t*-test (df = 46) | *t* = 0.6739 | | *t* = 0.2024 | | *t* = 0.7770 | | Z = 0.3381 | | *t* = 0.0166 | |
| | *p* = 0.5037 | | *p* = 0.8405 | | *p* = 0.4411 | | *p* = 0.7353 | | *p* = 0.9869 | |

*p*—significance level, A1—no stoma, A2—stoma, M—mean, Me—median, SD—standard deviation.

The next step of the analysis consisted in comparing the quality of sexual life as assessed using the FSFI questionnaire at 3 study time points: Before the surgery (Measurement I), six months after the surgery (Measurement II), and 12 months after the surgery (Measurement III). Detailed results in desire, arousal, lubrication, orgasm, satisfaction, and pain/discomfort subscales as well as the overall FSFI scores are presented in Table 3. At the first timepoint, no differences were observed in FSFI scores between the study groups ($p > 0.05$). Statistically significant differences in results were observed in Measurements II and III in the subscales of arousal (II:$p = 0.0068$, III:$p = 0.0018$), lubrication (II:$p = 0.0221$,

III:*p* = 0.0134), orgasm (II:*p* = 0.0044, III:*p* = 0.0021), satisfaction (II:*p* = 0.0021, III:*p* = 0.0433), and pain/discomfort (II:*p* = 0.0343, III:*p* = 0.0473). In all cases, lower results corresponding to lower quality of sexual functioning were observed in patients in whom stoma had been performed. The overall FSFI score was also significantly lower in the group of patients with stoma (II:*p* = 0.0118, III:*p* = 0.0025).

**Table 3.** FSFI scores in individual study groups.

| Variables Included in the Analysis | Group | M | Me | SD | Mann–Whitney's U-Test |
|---|---|---|---|---|---|
| I—Desire [Measurement I] | A1 | 4.85 | 4.80 | 1.05 | Z = 1.3502 *p* = 0.1770 |
| | A2 | 4.30 | 4.20 | 1.40 | |
| II—Arousal [Measurement I] | A1 | 4.54 | 5.70 | 2.13 | Z = 1.3427 *p* = 0.1794 |
| | A2 | 3.72 | 4.80 | 2.43 | |
| III—Lubrication [Measurement I] | A1 | 4.27 | 5.10 | 1.98 | Z = 0.3029 *p* = 0.7619 |
| | A2 | 3.65 | 5.10 | 2.51 | |
| IV—Orgasm [Measurement I] | A1 | 4.38 | 5.20 | 2.02 | Z = 1.4471 *p* = 0.1479 |
| | A2 | 3.39 | 4.80 | 2.49 | |
| V—Satisfaction [Measurement I] | A1 | 5.17 | 5.60 | 0.91 | Z = 0.3271 *p* = 0.7436 |
| | A2 | 4.75 | 5.60 | 1.45 | |
| VI—Pain/discomfort [Measurement I] | A1 | 4.16 | 4.80 | 2.10 | Z = 1.1906 *p* = 0.2338 |
| | A2 | 3.32 | 4.00 | 2.42 | |
| Overall FSFI score [Measurement I] | A1 | 27.37 | 30.90 | 9.36 | Z = 1.4967 *p* = 0.1345 |
| | A2 | 23.13 | 28.70 | 11.62 | |
| I—Desire [Measurement II] | A1 | 4.66 | 6.00 | 1.68 | Z = 1.9151 *p* = 0.0555 |
| | A2 | 3.83 | 4.20 | 1.47 | |
| II—Arousal [Measurement II] | A1 | 3.79 | 5.10 | 2.48 | Z = 2.7056 *p* = 0.0068 |
| | A2 | 1.96 | 2.10 | 2.06 | |
| III—Lubrication [Measurement II] | A1 | 3.64 | 5.10 | 2.54 | Z = 2.2882 *p* = 0.0221 |
| | A2 | 1.80 | 0.00 | 2.41 | |
| IV—Orgasm [Measurement II] | A1 | 3.94 | 5.60 | 2.70 | Z = 2.8509 *p* = 0.0044 |
| | A2 | 1.70 | 0.00 | 2.36 | |
| V—Satisfaction [Measurement II] | A1 | 4.83 | 5.20 | 1.10 | Z = 3.0704 *p* = 0.0021 |
| | A2 | 3.63 | 3.20 | 1.34 | |
| VI—Pain/discomfort [Measurement II] | A1 | 3.66 | 4.80 | 2.55 | Z = 2.1160 *p* = 0.0343 |
| | A2 | 2.00 | 0.00 | 2.51 | |
| Overall FSFI score [Measurement II] | A1 | 24.52 | 32.80 | 12.69 | Z = 2.5187 *p* = 0.0118 |
| | A2 | 14.93 | 9.50 | 10.86 | |
| I—Desire [Measurement III] | A1 | 5.26 | 6.00 | 1.41 | Z = 1.8930 *p* = 0.0584 |
| | A2 | 4.62 | 4.80 | 1.58 | |
| II—Arousal [Measurement III] | A1 | 4.70 | 6.00 | 2.24 | Z = 3.1205 *p* = 0.0018 |
| | A2 | 2.71 | 2.70 | 2.48 | |
| III—Lubrication [Measurement III] | A1 | 4.46 | 5.40 | 2.15 | Z = 2.4730 *p* = 0.0134 |
| | A2 | 2.69 | 2.40 | 2.59 | |
| IV—Orgasm [Measurement III] | A1 | 4.69 | 6.00 | 2.23 | Z = 3.0707 *p* = 0.0021 |
| | A2 | 2.64 | 2.40 | 2.56 | |

**Table 3.** *Cont.*

| Variables Included in the Analysis | Group | M | Me | SD | Mann–Whitney's U-Test |
|---|---|---|---|---|---|
| V—Satisfaction [Measurement III] | A1 | 5.18 | 5.60 | 1.21 | Z = 2.0210 |
| | A2 | 4.28 | 4.80 | 1.53 | p = 0.0433 |
| VI—Pain/discomfort [Measurement III] | A1 | 4.70 | 5.60 | 2.18 | Z = 1.9836 |
| | A2 | 2.97 | 3.20 | 2.80 | p = 0.0473 |
| Overall FSFI score [Measurement III] | A1 | 29.00 | 34.90 | 10.99 | Z = 3.0268 |
| | A2 | 19.91 | 18.70 | 12.76 | p = 0.0025 |

*p*—significance level, A1—no stoma, A2—stoma, M—mean, Me—median, SD—standard deviation.

The next stage of statistical analysis focused on the sexual functioning of patients in both groups as assessed at individual time points. Within group A1, significant differences were observed within the subscales of arousal ($p = 0.0478$) and pain/discomfort ($p = 0.0191$) as well as in the overall FSFI scores ($p = 0.0243$). Findings included an initial decrease in the results at the second measurement timepoint and a significant increase 12 months after the treatment. Notably, scores higher than those reported prior the procedure were observed in all cases at the third measurement timepoint. Within group A2, significant or highly significant differences were observed between individual timepoints with regard to the subscales of arousal ($p = 0.0032$), lubrication ($p = 0.0051$), and orgasm ($p = 0.0109$), as well as to the overall FSFI score ($p = 0.0142$). Significant drops were observed in these between the Measurement I and Measurement II timepoints. At Measurement III, the scores were on an upward trend; however, they remained much lower than those at the baseline. Details are presented in Table 4.

The next stage of the statistical analysis consisted in the analysis of correlations between the overall FSFI score and the demographic and clinical data. The relationship between the overall FSFI score and the quantitative demographic variables of weight, height, BMI, parity, duration of hospital stay was negligible ($p > 0.05$) at each of the measurement timepoints. Similarly, FSFI results did not differ significantly ($p > 0.05$) in relation to qualitative variables such as the incidence of postoperative complications, type of adjuvant treatment, cancer staging, educational background, area of residence, employment status, or socioeconomic status.

Highly significant correlation was observed at Measurements I ($p = 0.0238$) and III ($p = 0.0084$) between the overall assessment of the quality of sexual life and the type of surgical procedure. Lower results corresponding to worse self-assessment of sexual functioning were observed for procedures involving enterostomy formation. No statistical significance was observed between the results of anterior resection laparoscopic anterior resection procedures. No statistically significant differences in the results were observed for individual types of surgical procedures ($p > 0.05$) at Measurement II. Irrespective of the measurement timepoint, the lowest results were observed for abdominoperineal resection of rectum.

The overall FSFI scores at Measurement I did not differ significantly ($p > 0.05$) for individual modalities of neoadjuvant treatment, whereas the differences at Measurements II ($p = 0.0149$) and III ($p = 0.0433$) were significant. The lowest results were observed for neoadjuvant radiotherapy.

A highly significant difference in results ($p < 0.01$) was also observed for the variable of marital status—regardless of the measurement timepoint, better sexual functioning was reported by married patients as compared to patients in partnership-based relationships.

Irrespective of the study group, a statistically significant difference ($p = 0.0208$) was observed for the variable of age at the Measurement II timepoint. This was a negative and low-level correlation. Details are presented in Tables 5 and 6.

**Table 4.** FSFI scores at individual time points.

| Group | Variables Included in the Analysis | | M | Me | SD | Friedman's Test |
|---|---|---|---|---|---|---|
| No stoma (A1) | I—Desire | [Measurement I] | 4.85 | 4.80 | 1.05 | T = 5.9385 p = 0.0513 |
| | | [Measurement II] | 4.66 | 6.00 | 1.68 | |
| | | [Measurement III] | 5.26 | 6.00 | 1.41 | |
| | II—Arousal | [Measurement I] | 4.54 | 5.70 | 2.13 | T = 6.08 p = 0.0478 |
| | | [Measurement II] | 3.79 | 5.10 | 2.48 | |
| | | [Measurement III] | 4.70 | 6.00 | 2.24 | |
| | III—Lubrication | [Measurement I] | 4.27 | 5.10 | 1.98 | T = 3.4865 p = 0.175 |
| | | [Measurement II] | 3.64 | 5.10 | 2.54 | |
| | | [Measurement III] | 4.46 | 5.40 | 2.15 | |
| | IV—Orgasm | [Measurement I] | 4.38 | 5.20 | 2.02 | T = 4.9143 p = 0.0857 |
| | | [Measurement II] | 3.94 | 5.60 | 2.70 | |
| | | [Measurement III] | 4.69 | 6.00 | 2.23 | |
| | V—Satisfaction | [Measurement I] | 5.17 | 5.60 | 0.91 | T = 4.0256 p = 0.1336 |
| | | [Measurement II] | 4.83 | 5.20 | 1.10 | |
| | | [Measurement III] | 5.18 | 5.60 | 1.21 | |
| | VI—Pain/discomfort | [Measurement I] | 4.16 | 4.80 | 2.10 | T = 7.9143 p = 0.0191 |
| | | [Measurement II] | 3.66 | 4.80 | 2.55 | |
| | | [Measurement III] | 4.70 | 5.60 | 2.18 | |
| | Overall FSFI score | [Measurement I] | 27.37 | 30.90 | 9.36 | T = 7.4382 p = 0.0243 |
| | | [Measurement II] | 24.52 | 32.80 | 12.69 | |
| | | [Measurement III] | 29.00 | 34.90 | 10.99 | |
| Stoma (A2) | I—Desire | [Measurement I] | 4.30 | 4.20 | 1.40 | T = 4.6944 p = 0.0956 |
| | | [Measurement II] | 3.83 | 4.20 | 1.47 | |
| | | [Measurement III] | 4.62 | 4.80 | 1.58 | |
| | II—Arousal | [Measurement I] | 3.72 | 4.80 | 2.43 | T = 11.5143 p = 0.0032 |
| | | [Measurement II] | 1.96 | 2.10 | 2.06 | |
| | | [Measurement III] | 2.71 | 2.70 | 2.48 | |
| | III—Lubrication | [Measurement I] | 3.65 | 5.10 | 2.51 | T = 10.5538 p = 0.0051 |
| | | [Measurement II] | 1.80 | 0.00 | 2.41 | |
| | | [Measurement III] | 2.69 | 2.40 | 2.59 | |
| | IV—Orgasm | [Measurement I] | 3.39 | 4.80 | 2.49 | T = 9.0313 p = 0.0109 |
| | | [Measurement II] | 1.70 | 0.00 | 2.36 | |
| | | [Measurement III] | 2.64 | 2.40 | 2.56 | |
| | V—Satisfaction | [Measurement I] | 4.75 | 5.60 | 1.45 | T = 5.8611 p = 0.0534 |
| | | [Measurement II] | 3.63 | 3.20 | 1.34 | |
| | | [Measurement III] | 4.28 | 4.80 | 1.53 | |
| | VI—Pain/discomfort | [Measurement I] | 3.32 | 4.00 | 2.42 | T = 5.7288 p = 0.057 |
| | | [Measurement II] | 2.00 | 0.00 | 2.51 | |
| | | [Measurement III] | 2.97 | 3.20 | 2.80 | |

**Table 4.** *Cont.*

| Group | Variables Included in the Analysis | M | Me | SD | Friedman's Test |
|---|---|---|---|---|---|
| Overall FSFI score | [Measurement I] | 23.13 | 28.70 | 11.62 | T = 8.5122 $p = 0.0142$ |
| | [Measurement II] | 14.93 | 9.50 | 10.86 | |
| | [Measurement III] | 19.91 | 18.70 | 12.76 | |

*p*—significance level, A1—no stoma, A2—stoma, M—mean, Me—median, SD—standard deviation.

**Table 5.** Correlation between FSFI scores and quantitative clinical and demographic stales.

| | Variables Included in the Analysis | r | p |
|---|---|---|---|
| Weight | FSFI [Measurement I] | −0.0052 | 0.9721 |
| | FSFI [Measurement II] | 0.1164 | 0.4309 |
| | FSFI [Measurement III] | 0.0104 | 0.9440 |
| Height | FSFI [Measurement I] | 0.1487 | 0.3132 |
| | FSFI [Measurement II] | 0.0748 | 0.6132 |
| | FSFI [Measurement III] | 0.0803 | 0.5877 |
| BMI | FSFI [Measurement I] | −0.0920 | 0.5339 |
| | FSFI [Measurement II] | 0.1746 | 0.2352 |
| | FSFI [Measurement III] | 0.0368 | 0.8037 |
| Hospitalization time | FSFI [Measurement I] | −0.1914 | 0.1924 |
| | FSFI [Measurement II] | 0.0277 | 0.8518 |
| | FSFI [Measurement III] | 0.0794 | 0.5918 |
| Parity | FSFI [Measurement I] | 0.0981 | 0.5070 |
| | FSFI [Measurement II] | 0.1149 | 0.4367 |
| | FSFI [Measurement III] | 0.0809 | 0.5847 |
| Age | FSFI [Measurement I] | −0.1521 | 0.3020 |
| | FSFI [Measurement II] | −0.3329 | 0.0208 |
| | FSFI [Measurement III] | −0.1246 | 0.3987 |

*p*—significance level.

**Table 6.** Correlation between FSFI scores and qualitative clinical and demographic stales.

| Variables Included in the Analysis | Data Filter | M | Me | SD | Mann–Whitney's U-Test\Kruskall-Wallis Test |
|---|---|---|---|---|---|
| Type of surgery | | | | | |
| FSFI [Measurement I] | Anterior resection | 28.09 | 31.50 | 8.89 | H = 9.4569 $p = 0.0238$ |
| | Laparoscopic anterior resection | 26.45 | 30.80 | 10.30 | |
| | abdominoperineal resection | 16.13 | 7.50 | 12.23 | |
| | Hartmann's procedure | 30.77 | 31.70 | 3.11 | |
| FSFI [Measurement II] | Anterior resection | 24.64 | 32.80 | 12.44 | H = 6.6092 $p = 0.0855$ |
| | Laparoscopic anterior resection | 24.35 | 32.90 | 13.61 | |
| | abdominoperineal resection | 15.79 | 8.35 | 12.89 | |
| | Hartmann's procedure | 13.99 | 9.50 | 8.66 | |

**Table 6.** *Cont.*

| Variables Included in the Analysis | Data Filter | M | Me | SD | Mann–Whitney's U-Test\Kruskall-Wallis Test |
|---|---|---|---|---|---|
| FSFI [Measurement III] | Anterior resection | 30.11 | 35.35 | 10.70 | H = 11.7098 *p* = 0.0084 |
| | Laparoscopic anterior resection | 27.58 | 34.20 | 11.70 | |
| | abdominoperineal resection | 15.32 | 7.80 | 12.41 | |
| | Hartmann's procedure | 24.93 | 30.80 | 11.67 | |
| Neoadjuvant treatment | | | | | |
| FSFI [Measurement I] | none | 25.90 | 30.40 | 11.24 | H = 2.1982 *p* = 0.5323 |
| | radiotherapy | 22.55 | 28.30 | 11.72 | |
| | chemotherapy | 29.10 | 29.60 | 2.88 | |
| | radiochemotherapy | 27.22 | 31.90 | 9.94 | |
| FSFI [Measurement II] | none | 24.11 | 32.80 | 12.51 | H = 10.4742 *p* = 0.0149 |
| | radiotherapy | 12.11 | 7.20 | 10.79 | |
| | chemotherapy | 18.60 | 18.70 | 0.56 | |
| | radiochemotherapy | 25.86 | 32.80 | 11.97 | |
| FSFI [Measurement III] | none | 25.34 | 34.90 | 15.08 | H = 8.1351 *p* = 0.0433 |
| | radiotherapy | 18.64 | 14.45 | 13.02 | |
| | chemotherapy | 22.50 | 18.70 | 7.20 | |
| | radiochemotherapy | 30.66 | 34.85 | 8.89 | |
| Adjuvant treatment | | | | | |
| FSFI [Measurement I] | none | 25.04 | 29.80 | 10.91 | H = 0.3516 *p* = 0.8388 |
| | chemotherapy | 25.14 | 30.75 | 11.83 | |
| | radiochemotherapy | 28.33 | 27.80 | 2.82 | |
| FSFI [Measurement II] | none | 21.61 | 27.35 | 13.03 | H = 2.6381 *p* = 0.2674 |
| | chemotherapy | 16.53 | 8.20 | 13.42 | |
| | radiochemotherapy | 16.60 | 18.35 | 4.03 | |
| FSFI [Measurement III] | none | 24.70 | 31.95 | 13.13 | H = 0.7292 *p* = 0.6945 |
| | chemotherapy | 24.18 | 30.15 | 13.14 | |
| | radiochemotherapy | 25.60 | 24.75 | 8.54 | |
| Post-surgical complications | | | | | |
| FSFI [Measurement I] | none | 25.94 | 30.70 | 10.07 | Z = 0.5549 *p* = 0.5790 |
| | observed | 22.72 | 28.70 | 13.05 | |
| FSFI [Measurement II] | none | 18.84 | 18.00 | 12.55 | Z = 0.9552 *p* = 0.3395 |
| | observed | 24.60 | 32.80 | 12.91 | |
| FSFI [Measurement III] | none | 23.46 | 30.80 | 12.99 | Z = 1.2429 *p* = 0.2139 |
| | observed | 29.77 | 32.10 | 9.78 | |

**Table 6.** *Cont.*

| Variables Included in the Analysis | Data Filter | M | Me | SD | Mann–Whitney's U-Test\Kruskall-Wallis Test |
|---|---|---|---|---|---|
| | | | | | |
| | | Disease staging (I–III) | | | |
| FSFI [Measurement I] | I | 24.69 | 29.10 | 11.04 | H = 3.2311 $p$ = 0.3573 |
| | IIA | 25.82 | 30.00 | 10.21 | |
| | IIIB | 28.55 | 32.90 | 11.46 | |
| | IIIC | 25.45 | 29.95 | 10.09 | |
| FSFI [Measurement II] | I | 23.69 | 32.85 | 13.53 | H = 5.9091 $p$ = 0.1161 |
| | IIA | 18.08 | 18.00 | 11.71 | |
| | IIIB | 24.30 | 32.80 | 13.31 | |
| | IIIC | 13.47 | 7.80 | 11.39 | |
| FSFI [Measurement III] | I | 25.23 | 31.05 | 11.94 | H = 0.4719 $p$ = 0.9250 |
| | IIA | 23.40 | 31.10 | 13.18 | |
| | IIIB | 24.37 | 32.45 | 15.07 | |
| | IIIC | 26.58 | 35.05 | 14.08 | |
| | | Educational background | | | |
| FSFI [Measurement I] | elementary | 21.10 | 26.00 | 11.05 | H = 3.6209 $p$ = 0.3054 |
| | vocational | 27.64 | 30.85 | 8.37 | |
| | secondary | 26.42 | 31.30 | 10.70 | |
| | higher | 23.45 | 30.15 | 12.21 | |
| FSFI [Measurement II] | elementary | 11.48 | 8.60 | 7.75 | H = 5.9592 $p$ = 0.1136 |
| | vocational | 25.56 | 33.00 | 12.54 | |
| | secondary | 21.20 | 23.50 | 12.66 | |
| | higher | 17.42 | 9.00 | 13.70 | |
| FSFI [Measurement III] | elementary | 22.65 | 24.90 | 12.81 | H = 0.5749 $p$ = 0.9021 |
| | vocational | 26.05 | 31.45 | 12.28 | |
| | secondary | 24.87 | 31.10 | 12.86 | |
| | higher | 24.18 | 32.70 | 13.93 | |
| | | Area of residence | | | |
| FSFI [Measurement I] | urban | 25.62 | 30.70 | 10.64 | Z = 0.5338 $p$ = 0.5935 |
| | rural | 24.59 | 29.50 | 10.92 | |
| FSFI [Measurement II] | urban | 20.14 | 19.10 | 12.63 | Z = 0.1515 $p$ = 0.8796 |
| | rural | 19.35 | 10.60 | 13.33 | |
| FSFI [Measurement III] | urban | 23.69 | 30.30 | 12.86 | Z = 0.4877 $p$ = 0.6257 |
| | rural | 27.23 | 32.90 | 12.01 | |

**Table 6.** *Cont.*

| Variables Included in the Analysis | Data Filter | M | Me | SD | Mann–Whitney's U-Test\Kruskall-Wallis Test |
|---|---|---|---|---|---|
| | **Employment status** | | | | |
| FSFI [Measurement I] | retired/on disablement pension | 24.58 | 29.60 | 10.92 | H = 1.3071 $p = 0.5202$ |
| | regular employment/company owner | 27.02 | 31.25 | 10.61 | |
| | housekeeping | 30.60 | 30.60 | 1.56 | |
| FSFI [Measurement II] | retired/on disablement pension | 18.17 | 10.00 | 12.83 | H = 2.9775 $p = 0.2257$ |
| | regular employment/company owner | 25.02 | 28.40 | 11.88 | |
| | housekeeping | 25.95 | 25.95 | 9.69 | |
| FSFI [Measurement III] | retired/on disablement pension | 25.33 | 31.15 | 12.24 | H = 1.8582 $p = 0.3949$ |
| | regular employment/company owner | 20.51 | 24.15 | 14.49 | |
| | housekeeping | 33.05 | 33.05 | 3.18 | |
| | **Marital status** | | | | |
| FSFI [Measurement I] | married | 27.76 | 31.75 | 9.15 | Z = 2.9533 $p = 0.0031$ |
| | in partnership | 18.07 | 17.35 | 11.74 | |
| FSFI [Measurement II] | married | 24.04 | 32.00 | 11.89 | Z = 3.8386 $p = 0.0001$ |
| | in partnership | 7.58 | 6.40 | 3.96 | |
| FSFI [Measurement III] | married | 27.20 | 33.55 | 12.12 | Z = 2.5981 $p = 0.0094$ |
| | in partnership | 16.98 | 14.45 | 11.25 | |
| | **Socioeconomic status** | | | | |
| FSFI [Measurement I] | average | 22.67 | 27.10 | 11.32 | H = 4.3520 $p = 0.1135$ |
| | good | 29.04 | 31.95 | 8.15 | |
| | very good | 19.70 | 19.70 | 20.51 | |
| FSFI [Measurement II] | average | 16.05 | 9.50 | 11.97 | H = 4.3976 $p = 0.1109$ |
| | good | 22.84 | 28.40 | 12.74 | |
| | very good | 32.70 | 32.70 | 2.12 | |
| FSFI [Measurement III] | average | 21.61 | 27.60 | 12.89 | H = 3.0044 $p = 0.2226$ |
| | good | 26.90 | 31.50 | 12.20 | |
| | very good | 34.80 | 34.80 | 0.85 | |

*p*—significance level, M—mean, Me—median, SD—standard deviation.

## 4. Discussion

This paper assesses the demographic and clinical factors that influence the self-assessment of the quality of sexual life in female patients receiving surgical treatment due to colorectal cancer. The study population was divided into two groups, namely patients requiring stoma formation (A2) and patients in whom gastrointestinal continuity was maintained (A1). Study variables were measured using the international standardized Female Sexual Index Function (FSFI) questionnaire. Differences in sexual satisfaction were analyzed in relation to the study groups, the measurement timepoints (preoperative, six and 12 months after the surgery), as well as the clinical and demographic factors.

Estimates show that up to 75% of patients treated for colorectal cancer experience sexual functioning disorders with nearly 1/3 declaring complete temporary of permanent discontinuation of intercourses [6]. As shown by the results of our studies, the

self-assessment of the quality of sexual life is significantly worse among women subjected to surgical procedures requiring stoma formation; it must be noted that no significant differences had been observed between the groups in any of the FSFI subscales at the first measurement timepoint ($p$ >0.05). Other authors confirm the negative impact of stoma on one's own body image and thus on the worsening of sexual functioning [7,8]. Negative perception of the altered physical image may be a predictor of distress and depressive disorders [9]. Medical professionals may contribute to the reduction of sexual disorders in patients with stoma by means of education regarding appropriate hygiene (reduction of odor, skin irritation, management of waste) as well as regarding the common prevalence of these problems [10]. While the surgery has no impact on degree of sexual desire, patients with stoma may present with anxiety regarding their partner's reaction to their altered physicality or regarding possible leaks from stoma bags during sexual activity; the patients should be therefore instructed to empty their bags prior to the intercourse [11]. The correlation between stoma and worsened sexual functioning was also described in other studies [7,11–14].

Another aspect of the statistical analysis consisted in the comparison of the quality of sexual functioning as self-assessed by patients in each group depending on the study timepoint. Findings in both study groups included an initial decrease in the results at the second measurement timepoint and a significant increase 12 months after the treatment. Scores higher that those reported prior to the surgery were observed in the no stoma group of patients 12 months after the treatment. In the stoma group, the scores at Measurement III were already on an upward trend while remaining much lower than those at the baseline. According to other authors, patients not reporting any sexual dysfunctions prior to oncological treatment may experience changes in sexual functioning during or after cancer therapy [15]. Long-term results obtained by Zutshi et al. in 260 colorectal cancer patients also suggest a significant drop in sexual functioning within one year after the surgery [16].

In this study, a statistically significant difference was observed in the self-assessed sexual satisfaction depending on the type of neoadjuvant treatment. Neoadjuvant radiotherapy has significantly contributed to the worsening of sexual functioning as self-assessed six and 12 months after the surgery. Svanström Röjvall and numerous other researchers point at the negative consequences of preoperative irradiation on the subsequent sexual activity. Neoadjuvant radiotherapy contributes to vaginal dryness and induces menopause in premenopausal women [2,14,17,18]. Decreased androgen production, similar to that observed following gonadal resection, was also observed in other studies in women with pelvic cancer and no ovarian resection [4]. Traa et al. demonstrated that neoadjuvant radiotherapy, stoma, older age, and incidence of postoperative complications are associated with higher risk of sexual dysfunctions [13,14]. Similar observations were made by other authors [11].

In our study, patient's marital status was a differentiating factor in the self-assessment of sexual functioning in women subjected to surgical treatment of colorectal cancer. In the course of the statistical analysis, marries patients were compared against patients in partnerships. As shown in the review by Wezel et al., the incidence of sexual dysfunctions was correlated with marital status and radiation dose applied (>50.4 Gy) [19]. In our study, a relationship between patient's age and the incidence of sexual dysfunctions three months after the surgery was also demonstrated irrespective of the type of surgical procedure. No statistically significant differences were observed one year after the surgery. As noted by Traa et al., older age is a risk factor for sexual dysfunctions [13,14]. No impact of sociodemographic status on the quality of sexual life was observed in our study. Other authors stress that health care professionals should provide particular support to patients with low sociodemographic status by initiating conversations on sexual life in the period between disease diagnosis and the end of cancer treatment [20].

The results presented herein contribute to understanding the causes of sexual dysfunctions in female patients undergoing surgeries due to colorectal cancer and provide

instigation for interventional studies involving the elements of psychophysical rehabilitation. An added scientific value consists in the use of a standardized, international assessment tool dedicated to multi-dimensional evaluation of sexual functioning in women as well as in the prospective character of the study. It is worth noting that the available literature consists mainly of retrospective date from small, primarily male samples. The need for further prospective, long-term studies and development of systemic solutions aimed at reducing sexual dysfunctions among women following colorectal cancer therapy was also pointed out by other authors [1].

We recognize our study does have certain limitations, including the relatively small sample size and lack of subject randomization.

### 5. Conclusions

1.  The quality of their sexual live as self-assessed by patients with stoma was significantly worse than that in patients in whom gastrointestinal tract continuity had been maintained.
2.  Irrespective of the study group, deterioration in the quality of sexual life was observed six months after the treatment. One year after the surgery, the results were showing an upward trend; however, they remained lower than the baseline values in patients with stomia while exceeding the baseline scores in all subscales in patients in whom no stoma was required.
3.  Marital status and age were the demographic variables responsible for significant differentiation of satisfaction with sexual life. Better quality of sexual functioning was reported by younger and/or married patients.
4.  Neoadjuvant radiotherapy was the clinical variable responsible for significant differentiation of satisfaction with sexual life. Worse quality of life results related to sexual functioning were reported by patients who had received neoadjuvant radiotherapy prior to the surgery.

**Author Contributions:** Conceptualization, M.T. (Magdalena Tarkowska); data curation, M.T. (Magdalena Tarkowska) and B.S.; formal analysis, M.T. (Magdalena Tarkowska), B.S., I.G.-M. and M.J.; investigation, M.T. (Magdalena Tarkowska) and T.N.; methodology, M.T. (Magdalena Tarkowska); project administration, M.T. (Magdalena Tarkowska); resources, I.G.-M., T.N., W.Z. and M.T. (Michał Tarkowski); software, M.T. (Magdalena Tarkowska), T.N., P.J. and M.T. (Michał Tarkowski); supervision, M.T. (Magdalena Tarkowska), W.Z. and P.J.; validation M.T. (Magdalena Tarkowska) and B.S.; visualization, M.T. (Magdalena Tarkowska); writing—original draft, M.T. (Magdalena Tarkowska); writing—review and editing, M.T. (Magdalena Tarkowska), W.Z. and I.G.-M. All authors have read and agreed to the published version of the manuscript.

**Funding:** This research received no external funding.

**Institutional Review Board Statement:** The conduct of the study was approved by the Bioethics Committee at the Nicolaus Copernicus University in Toruń (decision no. 283/2019)). All procedures performed in studies involving human participants were in accordance with the ethical standards of the institutional and/or national research committee and with the 1964 Helsinki declaration and its later amendments or comparable ethical standards.

**Informed Consent Statement:** Informed consed was obtained from all individuals patricipants included in a study.

**Data Availability Statement:** The datasets generated during and/or analysed Turing the current study are available from the corresponding author on responsablereques.

**Acknowledgments:** We wish to thank the patients and professional personnel in the Department of Surgical Oncology, Oncology Centre, Bydgoszcz, Poland, for their assistance in this study.

**Conflicts of Interest:** The authors declare no conflict of interest.

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
