# Peer review of "Sexual Functioning in Female Patients Undergoing Surgical Treatment for Colorectal Cancer—A Single-Center, Prospective Triple Timepoint Yearly Follow-Up"

_curroncol, doi:10.3390/curroncol29050269_

Round 1

Reviewer 1 Report

1- as is well explained in the results, women treated with abdominoperineal resection showed lower scores. In the same way women with stoma  ( after both hartmann and Miles procedure) showed lower scores. In my opinion it would be interesting, with a larger cohort of patients, to asess how having a permanent stoma (Miles procedure) or a temporary stoma ( Hartmann procedure) affects sexual life and behaviour.

Author Response

Dear Reviewer #1,

Thank you for reviewing our article titled ‘Sexual functioning in female patients undergoing surgical treatment for colorectal cancer – a single-center, prospective triple timepoint yearly follow-up’ We deeply appreciate your opinion as well as constructive comments that contributed to more profound consideration of issues addressed in our publication. The comments in your review will guide us in our future work.

            In response to those commentaries we clarified the Introduction section. We agree with you that could be interesting, with a larger cohort of patients, to asess how having a permanent stoma (Miles procedure) or a temporary stoma ( Hartmann procedure) affects sexual life and behaviour. We underlined in the limitation of tle study section, that our study sample is small, we did our research Turing pandemic time and many of patients withdrew from the study.

Thanks to the comments received. Your review, expecially a idea to compare sexual functioning of patients with Miles and Hartmann procedure will become an inspiration for our future research.

The Authors

Reviewer 2 Report

This is a very interesting approach into the colorectal cancer community. This study describe an important aspect of the human being the body image and sexual activity. It is a must to understand the way a person see themselves in order to view why its acting the way it does.

The study for future references and more scientific power should be repeated but using the randomization that lacked this time and a bigger sample size. Also it will be interesting to know what can be done for this patients to better themselves and perform sexually and body-image wise in a more positive manner. 

Regrettably, females in general have a lot of pressure to fit into a society of looks and sexuality wherever they go. It must be a bigger hassle for females as undergoing cancer and surgery and put more distress into their image and sexual behavior. This is why I encourage if possible, to help the more people as possible approaching with facts, and solutions for this population.   

Author Response

Dear Reviewer #2,

Thank you for reviewing our article titled ‘Sexual functioning in female patients undergoing surgical treatment for colorectal cancer – a single-center, prospective triple timepoint yearly follow-up’ We deeply appreciate your opinion as well as constructive comments that contributed to more profound consideration of issues addressed in our publication. The comments in your review will guide us in our future work.

            In response to those commentaries we clarified the Introduction section. We agree with you that females in general have a lot of pressure to fit into a society of looks and sexuality wherever they go. It must be a bigger hassle for females as undergoing cancer and surgery and put more distress into their image and sexual behavior. We underlined in the limitation of tle study section, that our study sample is small, without randomization . We did our research Turing pandemic time and many of patients withdrew from the study.

Thanks to the comments received. Your review will become an inspiration for our future research.

The Authors